# Chewing Behavior Attenuates Lung-Metastasis-Promoting Effects of Chronic Stress in Breast-Cancer Lung-Metastasis Model Mice

**DOI:** 10.3390/cancers14235950

**Published:** 2022-12-01

**Authors:** Jia-He Zhang, Ke-Yong Wang, Kin-Ya Kubo, Kagaku Azuma

**Affiliations:** 1Department of Anatomy, School of Medicine, University of Occupational and Environmental Health, 1-1 Iseigaoka, Yahatanishi-ku, Kitakyusyu 807-8555, Japan; 2Shared-Use Research Center, School of Medicine, University of Occupational and Environmental Health, 1-1 Iseigaoka, Yahatanishi-ku, Kitakyusyu 807-8555, Japan; 3Faculty of Human Life and Environmental Science, Nagoya Women’s University, 3-40 Shioji-cho, Mizuho-ku, Nagoya 467-8610, Japan

**Keywords:** psychological stress, chewing, breast cancer, metastasis, Glucocorticoid, β2-adrenergic receptor

## Abstract

**Simple Summary:**

Breast-cancer-related death is mainly caused by distal metastasis. One of the most common metastatic sites for breast cancer is the lung. In the mouse model of breast-cancer lung metastasis, we showed that chronic psychological stress accelerated the lung metastasis of breast cancer by increasing the level of stress hormones and their receptors, oxidative stress, and the subsequent signaling-molecules involving angiogenesis and matrix degradation. Chewing, or active mastication, is a practical behavior for coping with psychological stress effectively. We identified the fact that chewing behavior could relieve chronic stress and ameliorate the promoting effects of chronic psychological-stress on the lung metastasis of breast cancer, via modulating stress-hormones and their receptors, and the downstream signaling-pathways.

**Abstract:**

We assessed the effects of chewing behavior on the lung-metastasis-promoting impact of chronic psychological-stress in mice. Human breast-cancer cells (MDA-MB-231) were injected into the tail vein of female nude mice. Mice were randomly divided into stress, stress-with-chewing, and control groups. We created chronic stress by placing mice in small transparent tubes for 45 min, 3 times a day for 7 weeks. Mice in the stress-with-chewing group were allowed to chew wooden sticks during the experimental period. The histopathological examination showed that chronic psychological-stress increased lung metastasis, and chewing behavior attenuated the stress-related lung metastasis of breast-cancer cells. Chewing behavior decreased the elevated level of the serum corticosterone, normalized the increased expression of glucocorticoid, and attenuated the elevated expression of adrenergic receptors in lung tissues. We also found that chewing behavior normalized the elevated expression of inducible nitric oxide synthase, 4-hydroxynonenal, and superoxide dismutase 2 in lung tissues, induced by chronic stress. The present study demonstrated that chewing behavior could attenuate the promoting effects of chronic psychological-stress on the lung metastasis of breast-cancer cells, by regulating stress hormones and their receptors, and the downstream signaling-molecules, involving angiogenesis and oxidative stress.

## 1. Introduction

Breast cancer is currently the most commonly diagnosed cancer and has a high mortality among women worldwide [1]. Most breast-cancer-related deaths are caused by distant metastasis. The most common metastatic sites for breast cancers are the lung, bone, liver and brain. Lung metastasis is a complex multi-step process, influenced by genetic and environmental factors. Chronic psychological-stress is considered to be an accelerator of the progression and metastasis of breast cancer [2,3,4,5]. Exposure to chronic stress induces elevation in glucocorticoid and norepinephrine levels, through activation of the hypothalamic–pituitary–adrenal (HPA) axis and sympathetic nervous system (SNS). Both glucocorticoid and norepinephrine, through binding with their receptors, the glucocorticoid receptor (GR) and β2-adrenergic receptor (β2AR), respectively, facilitate breast-cancer metastasis through modulating various downstream-signaling pathways [4,6,7,8,9].

Both tumor necrosis factor-α (TNF-α) and transforming growth factor-β (TGF-β) are critical cytokines in the cancer microenvironment. Both TNF-α and TGF-β can induce and promote epithelial–mesenchymal transformation (EMT), angiogenesis and the metastatic process. Sustained and abnormal angiogenesis is considered to be critical for cancer metastasis [10]. Vascular endothelial growth factor (VEGF) is a key regulator of cancer angiogenesis [11,12]. TNF-α can modulate the inflammatory process, enhancing VEGF production. The degradation of the extracellular matrix (ECM) and the basement membrane is essential for cancer metastasis. Matrix metalloproteinases (MMPs) can promote the degradation of ECM and the basement membrane, accelerating metastasis [13].

Glucocorticoids could mediate effects on inducible nitric oxide synthase (iNOS), increasing NO-signaling in breast-cancer cells. Elevated expression of iNOS has been found to correlate with cancer progression, indicating that NO activation may drive cancer metastasis [14,15]. Oxidative stress is an essential regulator in cancer metastasis. The activation of GR or β2AR could enhance productivity of both the reactive nitrogen species (RNS) and reactive oxygen species (ROS). Four-hydroxynonenal (HNE) is a product of cellular lipid-peroxidation, and recognized as a formidable reactive-aldehyde. It is considered one of the major mediators of oxidative stress in cells and tissues. An increased level of HNE can induce molecular alterations of cancer cells, up-regulate EMT, and thus accelerate metastasis [16]. Superoxide dismutase 2 (SOD2) has a dual role in cancer progression and metastasis. SOD2 expression increases the cellular oxidant/antioxidant ratios, associated with cancer progression and metastasis. Increased SOD2-expression is associated with the metastatic progression of breast-cancer cells. The excessive expression of SOD2 in cancer cells is associated with increased expression of MMPs, inducing ECM degradation and metastasis progression [17,18].

Several healthy-lifestyle factors, such as physical activity and modest calorie-restriction, can improve the stress response [19,20]. Chewing behavior or active mastication could alleviate psychological-stress response [21,22,23]. We recently reported that chewing behavior can alleviate the promoting effects of chronic stress on the development and progression of breast cancer in model mice [24]. It is important to examine the impacts of chewing behavior on lung metastasis of breast-cancer cells under chronic stress conditions. The purpose of the present study was to evaluate the effects of chewing behavior on lung metastasis of breast-cancer cells and to unravel the underlying mechanisms in nude mice exposed to long-term restraint stress.

## 2. Materials and Methods

### 2.1. Cell Culture

The human-breast adenocarcinoma cell line MDA-MB-231 was purchased from the American Type Culture Collection (Manassas, VA, USA). Cells were maintained in a humidified incubator containing 5% CO_2_ at 37 °C, and cultured with the Dulbecco’s modified Eagle’s medium, containing 10% fetal bovine serum. The antibiotics, 1% penicillin and streptomycin, were added to prevent bacterial contamination [24].

### 2.2. Animal Models and Tail Vein Metastasis Assays

Female BALB/c nude mice (7 weeks old, *n* = 6/per group) were purchased from the Japan Charles River Laboratories (Hamamatsu, Shizuoka, Japan). All mice were fed with a commercial mouse diet (CE-2, CLEA Japan, Inc., Tokyo, Japan) and drinking water available ad libitum under specific pathogen-free (SPF) conditions (temperature: 23 ± 1 °C, humidity: 55 ± 5%, 12:12 h light/dark cycle). All animal experimental protocols in this study were reviewed and approved by the Ethics Committee for Animal Care and Experimentation of the University of Occupational and Environmental Health, Japan (Approval number: AE20-011, 24 August 2020).

After acclimation for 7 days, mice were then randomly distributed into control, stress, and stress-with-chewing groups (*n* = 6/group). The restraint and chewing-procedure was performed as previously described [25]. Animals in the stress and stress-with-chewing groups were placed in a transparent tube (inner diameter: 3.5 cm), in which the mice were able to move back and forth but not turn around, as previously reported [24]. Animals were placed in the tube for 45 min, 3 times a day for 7 weeks. Mice in the stress-with chewing group were allowed to chew a wooden stick (diameter: 2.0 mm) during the stress period. At the end of the experiment, the wooden sticks were checked, and all mice showed evidence of chewing behavior. Animals in the control group were neither exposed to stress nor given wooden sticks. On the first day of the stress exposure, 10 µL saline with 1 × 10^6^ MDA-MB-231 cells were injected intravenously via the tail vein [26].

### 2.3. Serum Corticosterone Assay

At the time of euthanasia, the animals were anesthetized by intraperitoneal injection, with a mixture of medetomidine, midazolam, and butorphanol. Blood samples were collected from the axillary artery from 10:00 to 11:00 a.m. on day 49 after cancer-cell inoculation. The serum was obtained by centrifugation of the blood sample at 3000 rpm for 15 min at 4 °C. The corticosterone level in the serum was measured using an enzyme-linked immunosorbent assay kit (Assaypro, St. Charles, MO, USA), following the manufacturer’s instructions [24].

### 2.4. Histopathological Image Analysis

Lung tissues (*n* = 5) were fixed in a 10% neutral buffered formalin solution. Five-micron-thick paraffin-embedded sections were prepared and processed for hematoxylin-eosin staining, and investigated under a light microscope (BX50, Olympus Corporation, Tokyo, Japan). Twenty sections per animal, with 5 fields per section were randomly selected to determine the number and size of the metastatic lung-nodules, using imaging software (CellSens, Olympus Corporation).

### 2.5. Immunohistochemical Staining

Paraffin sections were processed for deparaffinization with xylene and for rehydration with ethanol, and treated with an antigen-resuscitation solution (Dako, Santa Clara, CA, USA) in an autoclave for 15 min at 121 °C. After washing with PBS, sections were immersed in protein block, serum-free (Dako), for 15 min. Sections were incubated with diluted anti-glucocorticoid receptor (CST, Danvers, MA, USA, Table 1) and anti-β2AR (Abcam, Cambridge, UK, Table 1) at 4 °C overnight. Subsequently, sections were treated with biotinylated goat anti-rabbit IgG and streptavidin peroxidase complex (Nichirei Biosciences Inc., Tokyo, Japan) for 30 min. After staining with diaminobenzidine and counterstaining with hematoxylin, sections were observed and captured under a light microscope (VS120, Olympus Corporation) and a digital camera. To determine the percentage of GR- and β2AR-positive cells in the metastatic nodules, 5 sections per animal (*n* = 5) and 5 fields per section were randomly selected for evaluation. All measurements were conducted in a double-blind procedure, as previously reported [24].

### 2.6. Western Blot Analyses

Mouse lung-tissue homogenates, including metastatic nodules (*n* = 5/group) were lysed with radioimmunoprecipitation assay buffer (Millipore, Burlington, MA, USA) and then processed for centrifugation at 12,000 rpm for 30 min at 4 °C, and the supernatants then collected. The protein concentration was assessed and adjusted to 2 µg/µL, using a BCA protein assay kit (Thermo Fisher Scientific, Waltham, MA, USA). We used 4–12% Bis-Tris Gel (Thermo Fisher Scientific, Waltham, MA, USA) for electrophoresis and then transferred the protein concentration to PVDF membranes (Millipore, Bedford, MA, USA). Immunoblotting was performed with the primary antibodies summarized in Table 1, at 4 °C overnight. The PVDF membranes were incubated with a second antibody (1:3000, CST) for 60 min and the specific protein-bands were identified on the membranes, using an enhanced chemiluminescence kit (Cytiva, Buckinghamshire, UK). The bands were captured with an Ez-Capture MG system (Atto Corp., Tokyo, Japan) and analyzed with the Scion Image software (version 4.0.2, Scion Corp., Frederick, MD, USA). GAPDH was used as a loading control for target protein normalization. The uncropped blots are shown in Appendix A. Densitometry readings/intensity ratio of each band are shown in Appendix A.

### 2.7. Statistical Analysis

All experimental data are expressed as mean ± standard error. Differences between groups were statistically analyzed using GraphPad Prism version 7.03 (San Diego, CA, USA). All parameters were analyzed with one-way analysis of variance (ANOVA). Tukey–Kramer post hoc test was performed for the following test of one-way ANOVA. Differences were considered significant at a *p* value less than 0.05.

## 3. Results

### 3.1. Chewing Behavior Attenuated the Promoting Effects of Restraint Stress on Lung Metastasis

Animal body-weights were measured every week. We found that the body weight was significantly lower in the stress group from day 28 after cancer-cell inoculation, compared with the control group. From day 35 after inoculation, the body weight in the stress group was significantly lower than that in the stress-with-chewing group (Figure 1A). The lung metastases of breast-cancer cells were observed microscopically (Figure 1B). We found that the number and size of metastatic nodules in the stress group was significantly increased compared with those of the control and stress-with-chewing groups (Figure 1B–F).

### 3.2. Chewing Behavior Normalized Restraint Stress-Induced Elevation of GR and β2AR Expression, and Corticosterone Level

Immunohistochemical images showed that the immuno-positive cells of GR and β2AR in metastatic lung-tissues was higher in the stress group (Figure 2A,B). Compared with the control and stress-with-chewing groups, the percentage of GR- and β2AR-positive cells in the metastatic nodules of the stress group was significantly higher (*p* < 0.001, Figure 2C,D). Western blot analysis indicated that chronic restraint-stress induced a significant increase in the expression of both GR and β2AR in lung tissues, compared with the control group (Figure 2E,F). Compared with the stress group, the expression of GR was significantly reduced, and the expression of β2AR tended to be lower in the stress-with-chewing group (Figure 2E,F). Serum-corticosterone concentration in the stress group was significantly higher than that in the control and stress-with-chewing groups (*p* < 0.05, Figure 2G).

### 3.3. Chewing Behavior Normalized Restraint Stress-Induced Elevation of TNF-α, TGF-β, VEGF and MMP Expression

Western blot analysis showed that chronic restraint-stress caused elevated expression of TNF-α, TGF-β, VEGF, MMP2 and MMP9 in mouse lung-tissues. Compared with the stress group, the expression of TNF-α aνδ VEGF significantly decreased, and the expression of TGF-β, MMP2 and MMP9 tended to be lower in the stress-with-chewing group (Figure 3). These findings indicate that chronic restraint-stress promotes cancer metastasis through activating inflammatory-cytokines, angiogenesis, and extracellular-matrix remodeling.

### 3.4. Chewing Behavior Ameliorated Oxidative Stress Induced by Restraint Stress

Oxidative stress was assessed by determining the protein expression of iNOS, HNE and SOD2 in mouse lung-tissues. Chronic restraint-stress induced an elevated expression of iNOS, HNE, and SOD2 in lung tissues. The expression of HNE in the stress-with-chewing group tended to decrease, and there was no significant difference between the stress and stress-with-chewing groups. Chewing behavior during restraint-stress normalized the protein expression of iNOS and SOD2 in lung tissues (Figure 4). These findings suggest that chewing behavior during restraint-stress reduced oxidative-stress in lung tissues, including metastatic nodules.

## 4. Discussion

The major findings of the present study showed that chewing behavior during restraint stress reduced the promoting effect of sustained psychological stress on lung metastasis in a mouse model of breast-cancer cells. In line with the previous reports, long-term psychological stress stimulated the degradation of the matrix and promoted cancer angiogenesis, eventually accelerating the metastasis of the breast cancer [2,5,6,7]. Our results demonstrated that chronic restraint-stress induced elevation of the circulating-corticosterone level and GR expression in metastatic-cancer tissues. Glucocorticoids contain corticosterone in rodents, and cortisol, the main glucocorticoid in humans [27]. Glucocorticoids play important roles in various biological processes via binding to GR. Accumulating evidence suggests that an increased glucocorticoid level is associated with the progression and metastasis of breast cancer [8,28,29]. Persistent elevation of GR expression has a causal association with accelerated cancer-angiogenesis and metastasis [2,6]. Chewing behavior is an efficient approach for coping with stress through reducing elevated glucocorticoid-levels [22,24,27,30]. In this study, we found that chewing behavior during restraint stress significantly decreased the glucocorticoid level, and normalized the expression of GR in mouse lung-tissues. These findings suggest that chewing behavior could inhibit glucocorticoid elevation induced by psychological stress, and suppress cancer metastasis mediated by normalizing GR-expression.

Long-term psychological stress induced the release of epinephrine and norepinephrine from the adrenal gland and sympathetic-nerve terminals. The hyperfunction of epinephrine and norepinephrine drives cancer progression and metastasis by activating the receptors. The activation of β2AR-signaling accelerated angiogenesis and metastasis of breast cancer under chronic psychological-stress [5,31]. In this study, we found that chronic psychological-stress increased β2AR expression in the lung tissues. including metastatic nodules. Chewing behavior during chronic stress normalized the elevation of β2AR expression induced by chronic restraint-stress.

Many factors can affect the lung metastasis of breast cancer, including tumor angiogenesis. Sustained and abnormal angiogenesis is crucial for metastasis. VEGF and its receptor is one of the most important targets for cancer angiogenesis [11,32]. TNF-α and TGF-β are critical cytokines in the metastatic microenvironment, affecting cancer growth and metastasis. TNF-α could enhance the production of VEGF by modulating the inflammatory process and upregulating several genes associated with cancer invasion and metastasis [33]. TGF-β could disrupt lung capillary-walls and promote lung metastasis of breast-cancer cells [34]. MMPs are a group of zinc-dependent metalloenzymes able to degrade the extracellular matrix, promoting cancer invasion, angiogenesis and metastasis [35]. Both MMP2 and MMP9 are two special MMPs. The expression level of MMP2 and MMP9 was closely involved in breast-cancer metastasis [36]. MMP2 is the main proteolytic enzyme among MMPs, and is a main promoter of cancer-cell invasion and metastasis by degrading the basement membrane and accelerating distant metastasis. MMP9 could degrade ECM components and the basement membrane, playing a critical role in ECM remodeling and membrane cleavage. The overexpression of MMP9 was confirmed as intimately associated with cancer metastasis [37]. We consider that chewing behavior inhibits chronic stress-related lung metastasis through normalizing the increased expression of VEGF, TNF-α, TGF-β, MMP2 and MMP9.

Long-term psychological stress is associated with increased production of ROS and RNS. Recently, we found that chronic stress increased the expression of iNOS in breast-cancer cells [24]. The overexpression of iNOS can enhance cancer angiogenesis, progression, and metastasis mediated by ROS and RNS [14]. HNE, as an oxidative-stress marker, is a toxic product of lipid peroxidation. It was reported that HNE could elevate the expression of VEGF, promote angiogenesis and the metastasis of breast-cancer cells by modulating mitochondrial function [38]. Enhanced expression of SOD2 was associated with cancer metastatic cancer progression. SOD2 can induce an increase in cellular oxidant/antioxidant ratios, correlating directly with cancer angiogenesis, progression and metastasis. SOD2 overexpression in cancer cells could enhance the expression and function of MMPs, promoting matrix degradation and accelerating metastatic progression [17]. In this study, we found that chewing behavior normalized the elevation of iNOS, HNE and SOD2 expression in lung tissues.

Several potential limitations of this study should be mentioned. One limitation is that this study focused on in vivo experiment. By performing complementary in vitro experiments, we might offer more satisfactory conclusions. Another limitation is that we collected the *lung tissue* homogenates, including metastatic nodules and the lung parenchymal cells, for analyzing protein expression. Thus, the specific protein-producing cells cannot be fully detected. Nonetheless, this study offered an insight into the impacts of chewing on the lung-metastasis-promoting effects of chronic stress in the breast-cancer metastasis model.

## 5. Conclusions

This study demonstrated the fact that chewing behavior could ameliorate lung metastasis of the breast-cancer cells associated with chronic psychological-stress in mice, through modulating stress-hormones and their receptors, oxidative stress and the downstream signaling-molecules involving angiogenesis and matrix-degradation.

## Figures and Tables

**Figure 1 cancers-14-05950-f001:**
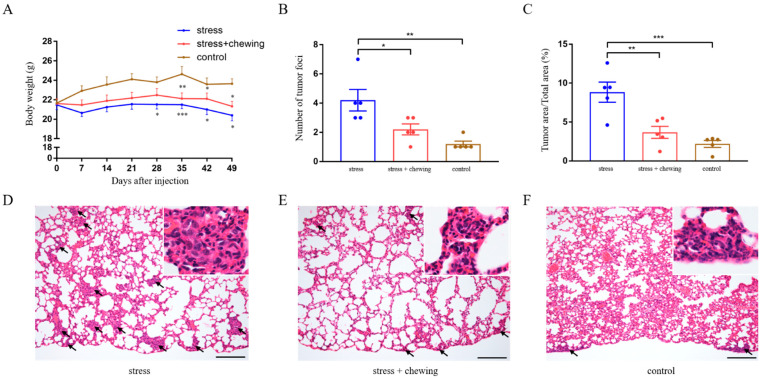
Chewing behavior attenuated the promoting effects of chronic stress on lung metastasis. Body-weight changes in the control, stress, and stress-with-chewing groups. * *p* < 0.05, ** *p* < 0.01, *** *p* < 0.001 vs. control group (**A**). The number of metastatic lung-nodules in the control, stress, and stress-with-chewing groups (**B**). The relative area of metastatic lung-nodules in the control, stress, and stress-with-chewing groups (**C**). The representative images of lung histology (**D**–**F**). Arrows indicate metastasis nodules. Scale bars = 200 μm. Data are expressed as mean ± SEM; *n* = 6/group; * *p* < 0.05, ** *p* < 0.01, *** *p* < 0.001.

**Figure 2 cancers-14-05950-f002:**
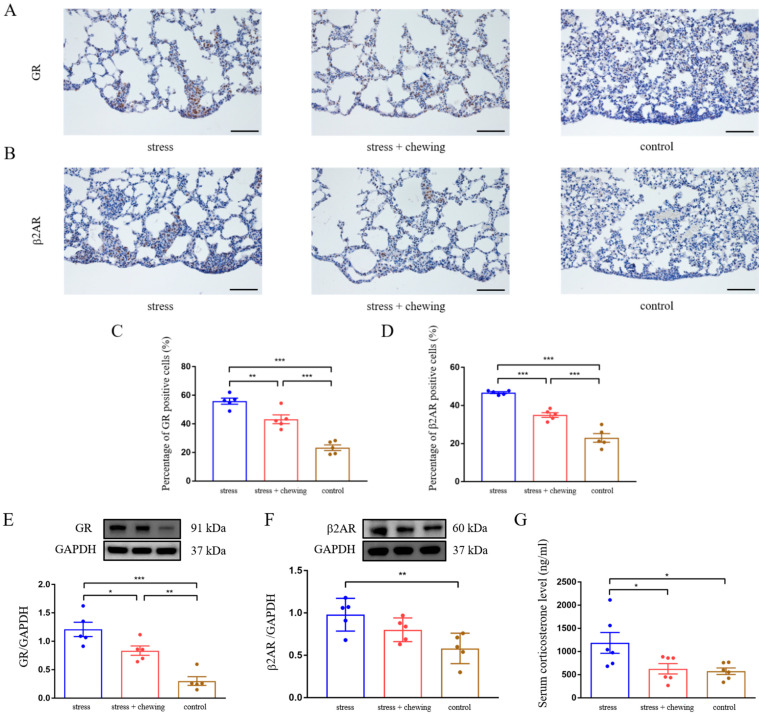
Chewing behavior normalized chronic-stress-induced elevation of GR and β2AR expression and corticosterone level. The representative immunostaining images of GR (**A**) and β2AR (**B**) in mouse lung-tissues. Scale bars = 100 μm. The percentage of GR- and β2AR-positive cells in mouse lung metastatic-nodules (**C**,**D**). The protein expression of GR (**E**) and β2AR (**F**) in mouse lung-tissues. Serum corticosterone levels in the control, stress, and stress-with-chewing groups (**G**). Data are expressed as mean ± SEM; *n* = 5/group; * *p* < 0.05, ** *p* < 0.01, *** *p* < 0.001.

**Figure 3 cancers-14-05950-f003:**
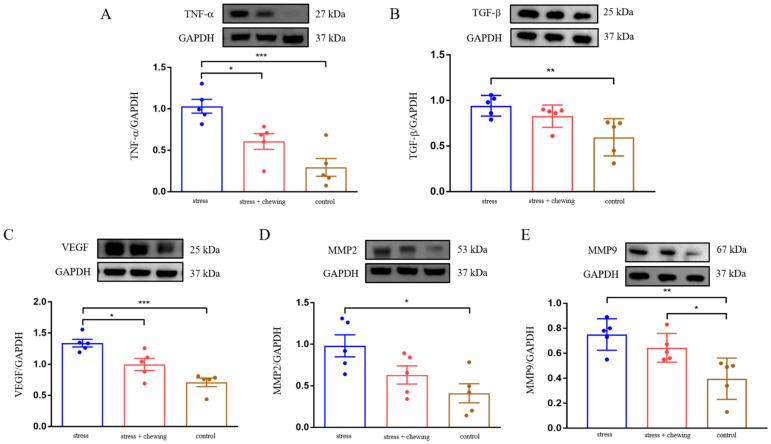
Chewing behavior normalized chronic-stress-induced elevation of TNF-α, TGF-β,VEGF and MMP expression. The protein expression of TNF-α (**A**), TGF-β (**B**) and VEGF (**C**), MMP2 (**D**) and MMP9 (**E**) in mouse lung-tissues. Data are expressed as mean ± SEM; *n* = 5/group; * *p* < 0.05, ** *p* < 0.01, *** *p* < 0.001.

**Figure 4 cancers-14-05950-f004:**
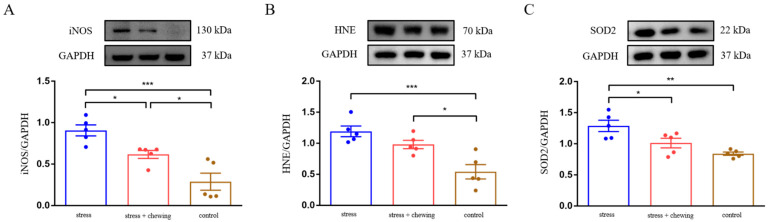
Chewing behavior ameliorated oxidative stress induced by chronic stress. The protein expression of iNOS (**A**), HNE (**B**), and SOD2 (**C**) in mouse lung-tissues. Data are expressed as mean ± SEM; *n* = 5/group; * *p* < 0.05, ** *p* < 0.01, *** *p* < 0.001.

**Table 1 cancers-14-05950-t001:** Antibodies used in this study.

Antibody	CAT#	Source	MW (kDa)	Dilution	Application
GR	#12041	CST	91	1:1000/1:400	WB/IHC
Β2AR	Ab135641	Abcam	60	1:100/1:50	WB/IHC
TNF-α	#L1120	SCBT	27	1:500	WB
TGF-β	#3711	CST	25	1:1000	WB
VEGF	GTX74091	GeneTex	25	1:1000	WB
MMP2	10373-2-AP	Proteintech	53	1:500	WB
MMP9	10375-2-AP	Proteintech	67	1:500	WB
iNOS	#2982	CST	130	1:1000	WB
HNE	MHN-100P	JaICA	70	16 μg/mL	WB
SOD2	#13141	CST	22	1:1000	WB
GAPDH	#2118	CST	37	1:1000	WB

MW, molecular weight; WB, Western blot; IHC, immunohistochemistry; CST, Cell Signaling Technology; SCBT, Santa Cruz Biotechnology; JaICA, Japan Institute for the Control of Aging.

## Data Availability

Data contained in the article and the original data that support the findings of the present study are available from the corresponding author upon reasonable request.

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
