# Peer review of "Chewing Behavior Attenuates Lung-Metastasis-Promoting Effects of Chronic Stress in Breast-Cancer Lung-Metastasis Model Mice"

_cancers, 2022, doi:10.3390/cancers14235950_

Round 1

Reviewer 1 Report

In this study authors evaluated the impact of chewing behavior on lung metastasis-promoting effects of chronic stress in breast cancer lung metastasis model mice. They showed that chewing behavior attenuates clinical signs stress, which further decrease tumor cells metastasis into the lung. Although the findings are interesting, further experiments/clarifications are needed before considering it for publication in Cancers.
1- In method section, please cite appropriate papers regarding the protocol for stress condition
2- Authors should clearly describe why they chose triple negative breast cancer cells and nude mice for this study? Do author expect to see similar results for other types of breast cancer and also in immunocompetent mice? This should be discussed
3- Quantification analysis should be provided for all IHC images.
4- Survival rate analysis should be provided.
5- Evaluation of chewing behavior lung metastasis-promoting effects of chronic stress in other cancer models will strengthen the paper.

Author Response

Thank you for the valuable comments. The response to the comments is as follows.

1- In method section, please cite appropriate papers regarding the protocol for stress condition.

Response: Thank you for your suggestion. We added one sentence and one reference in Materials and Methods section, ‘The restraint procedure was performed as previously described (Exp Gerontol 55:12-18, 2014)’.

2- Authors should clearly describe why they chose triple negative breast cancer cells and nude  

mice for this study? Do author expect to see similar results for other types of breast cancer and also in immunocompetent mice? This should be discussed

Response: Thank you for your indication. There are several kinds of breast cancer cell lines available. MDA-MB-231 is a human triple-negative breast cancer cell line. MCF-7 and T47-D are human breast cancer cell lines with estrogen, progesterone and glucocorticoid receptors, representing the luminal A subtype of breast cancer. 4T1 is mouse breast cancer cell line. In this study, we used MDA-MB-231 cells and nude mice to investigate lung metastasis. We would like to apply MCF-7, 4T1, and Balb/c, C57BL/6, transgenic, or nude mice to increase our understanding of the breast cancer in the near future.

3- Quantification analysis should be provided for all IHC images.

Response: We added the following sentences in the section of 2.5. Immunohistochemical staining. To calculate the percentage of GR- and b2AR-positive cells in the metastatic nodules, 5 sections per animal (n=5) and 5 fields per section, at 400× were randomly selected for evaluation. All measurements were conducted in a double-blind procedure as previously reported (Brain Sci 11:479, 2021).

4- Survival rate analysis should be provided.

Response: Thank you for this indication. Survival analysis is used to investigate the time it takes for an event of interest to occur, such as cancer studies for animal survival time analysis. Kaplan-Meier plots, Log-rank test and Cox proportional hazards regression are applied for survival analysis. In this study, we did not find any mice died during the experimental period. Therefore, we did not analyze the survival rate.

5- Evaluation of chewing behavior lung metastasis-promoting effects of chronic stress in other  

cancer models will strengthen the paper.

Response: Chewing is a practical behavior for coping with stress. Chewing could ameliorate osteoporosis and cognitive impairments induced by psychological stress (Exp Gerontol 15:12-18, 2014; Int J Mol Sci 21:5627, 2020). Recently, we found that chewing behavior could attenuate the tumor progression-enhancing effects of psychological stress in a breast cancer model (Brain Sci 11:479, 2021). We anticipate that chewing behavior could attenuate metastasis-promoting effects of chronic stress in other cancer models.

Reviewer 2 Report

The study of Dr Jia-He Zhang et al aims to delineate the protective effect of chewing on lung metastasis promoted by chronic stress, using an experimental mouse model. This study is in line with a previous report from the same group concerning the effect of chewing on the growth of human breast cancer cells subcutaneously xenografted in nude mice maintained under chronic psychological stress.

In the present study, experimental metastases were induced by tail vein injection of the MDA-MB-231 cells in 3 groups of animals: control group, chronic psychological stress induced by mouse restraint, and stress with chewing group.

The Authors report that chewing was associated with a lower level of serum corticosterone, a decreased number of lung metastatic nodules, a reduction in the level of some effectors of cancer cell growth and dissemination, including VEGF, MMP2, MMP9, and the normalization of the expression of glucocorticoid receptors (GR) and B2-adrenergic receptors (B2-AR).

The topic is of interest, but this study raises a series of issues.

The Authors did not provide any direct link between stress-induced corticosterone, epinephrine or norepinephrine and some features of cancer, such as proliferation and dissemination. The Authors could have performed complementary experiments in vitro to delineate the underlying mechanisms, they could have investigated proliferation and apoptosis in lung metastatic nodules, they could have used agonists and/ or antagonists of B2AR and GR in vitro and in vivo to delineate the respective impacts of these receptors.

The chronic stress condition was associated with a decrease in mouse body weight. Did the Authors notice any change in the dietary intake according to the different groups  ? In future studies, it might be interesting to quantify other hormones involved in cell proliferation, e.g. insulin, IGF-1, leptin, adiponectin.

The Authors did not mention whether psychological stress began from the day of injection or was delayed. This might be of importance, since stress hormones (e.g. adrenergic) might induce resistance of cancer cells to anoikis.

Chronic exposure of a receptor to its ligand leads to a desensitization process. Nevertheless, the Authors observed an increased accumulation of B2-AR and GR in metastatic nodules. This should be discussed.

It is not clear whether chewing interferes with tumor implantation or tumor growth. The Authors report a decrease in the number of the lung metastatic nodules in the stress + chewing group as compared with the stress group. Another interesting parameter would have been nodule size. Based on the histograms in Figure 1B and 1C, it seems that the size of the nodules are in the same range in the stress group and in the stress + chewing group. In contrast, the histological sections seem to display higher nodule sizes in the stress group.

It is not clear how many sections were analyzed to determine the number of metastatic nodules. In the Material and method section, it is mentioned line 120 "n=6", does it mean 2 mice/group or 6 mice/group ? And line 122: "Five fields were randomly selected from each section". Since panels 1B and 1C in Figure 1 display 5 dots/ group, does it mean that for each group only one section from one mouse was analyzed ? Due to the heterogeneity of tumor implantation, and the low number of metastatic nodules (mean value 1 to 4 nodules/ group), I am not confident with the representativeness.

According to the Scale bars in Figure 1D, 1E ,1F, the metastatic nodules are smaller than 100um. So it is not clear to me, what is the "lung metastatic tissue" which was analyzed by western blot. Is it pools of nodules ? in this case the Authors should describe the microdissection procedure. Is it homogenates from whole lungs ? in this case this is not metastatic tissue. This should be clearly stated to delineate which cell types might produce VEGF, iNOS, VEGF, TGFB and MMPs (although there is probably stromal cells infiltrating the metastatic nodules). If whole lung have been homogenized, it would have been interesting to compare the expression of these proteins i) in another tissue, e.g. liver, and ii) in lung from control (non metastatic) mice subjected to a psychological stress.

From a general point of view, the Authors should provide some enlargements of the histological and immunohistological micrographs. The enlargement in Fig.1F seems different from Fig.1D and 1E.

The original western blots (back matter) display a huge background, it is difficult to identify the main specific bands. This is also a problem for the interpretation of immunohistochemistry. As far as I understood, original western blots originate from 3 membranes. MMP9 (Fig.3E) and TGFB (Fig.3B) labeling have been performed sequentially on the same blot (regardless of the order), but the patterns are similar and thus the bands considered as specific are subject to caution. The expected size of B2-AR is higher than 46,000 (core protein), in the range 50,000-60,000 depending on glycosylation level. The immunoreactive band attributed to B2-AR in Fig.2D is probably below 40,000. Furthermore, its pattern of expression looks like a band of the same size on the iNOS blot (Fig.4A).  For TNF-a, another band of about 60,000 is also observed and displays the similar pattern of expression.

Glucococorticoids are potent immunomodulators. The Authors could consider in the future, alternative experimental models to immunodefficient mice, such as xenograft of mouse breast cancer 4T1 or EO771 cell lines in balb/c or C57BL/6 mice, respectively, or the use of transgenic mice or chemically induced carcinogenesis.

Author Response

Thank you for the valuable comments. The response to the comments is as follows.

  • The Authors did not provide any direct link between stress-induced corticosterone, epinephrine or norepinephrine and some features of cancer, such as proliferation and dissemination. The Authors could have performed complementary experiments in vitro to delineate the underlying mechanisms, they could have investigated proliferation and apoptosis in lung metastatic nodules, they could have used agonists and/ or antagonists of B2AR and GR in vitro and in vivo to delineate the respective impacts of these receptors.

Response: Thank you for your critical indication. In this study, we found that chewing alleviated stress and ameliorated the promoting effects of chronic stress on lung metastasis, via modulating corticosterone, epinephrine, norepinephrine and their receptors, and the downstream signaling pathways in vivo.

As you indicated, in order to delineate the detailed underlying mechanisms, it is necessary to perform experiments in vitro, investigating proliferation and apoptosis, or using agonists and antagonists of b2AR and GR.

According to the editor’s suggestion, we have to upload the revised file within 10 days. Therefore, we consider to do these experiments in our next study. We added a paragraph to mention the limitation of this study, including in vitro study, in the Discussion section.

  • The chronic stress condition was associated with a decrease in mouse body weight. Did the Authors notice any change in the dietary intake according to the different groups? In future studies, it might be interesting to quantify other hormones involved in cell proliferation, e.g. insulin, IGF-1, leptin, adiponectin.

Response: The time course of body weight changes (Fig. 1A) showed that there were no significant differences among the three groups from day 0 to 21 after cancer cell inoculation. The body weight in the stress group was significantly lower from day 28 after cancer cell inoculation. Unfortunately, we did not measure the daily dietary intake in this study. We would like to quantify the levels of insulin, IGF-1, leptin, adiponectin involved in cell proliferation in our next study.

  • The Authors did not mention whether psychological stress began from the day of injection or was delayed. This might be of importance, since stress hormones (e.g. adrenergic) might induce resistance of cancer cells to anoikis.

Response: Thank you for this indication. As to the cell inoculation time and the stress period, we established chronic restraint stress model for 7 weeks. The cancer cells were injected into the mice on the first day of the stress exposure, as previously reported (Brain Sci 11:479, 2021).

  • Chronic exposure of a receptor to its ligand leads to a desensitization process. Nevertheless, the Authors observed an increased accumulation of b2-AR and GR in metastatic nodules. This should be discussed.

Response: Chronic stress drives a sustained increase in glucocorticoid (GC) and norepinephrine levels via activating the hypothalamic-pituitary-adrenal (HPA) axis and sympathetic nervous system. These stress hormones bind to their receptors, GR and b2-AR, respectively. Upon ligand binding, GR translocates from the cytoplasm to the nucleus to regulate target gene expression, and downregulate the expression of GR itself. Chronic stress would deregulate or impair GR negative feedback. In addition to regulating its own gene expression, GR could modulate expression of corticotrophin-releasing hormone (CRH). We reported that chronic stress increased the circulating GC level, accompanied by decreased level of the central GR expression (Int J Mol Sci 21:5627, 2020). Some studies showed that under chronic stress, GC levels were high and the intracellular GR levels were increased (J Endocrinol 202:87-97, 2009). The inconsistencies may be due to variability in timing of assessment during the course of the chronic stress and poor reliability of measurement across studies (Psychol Bull 133:25-45, 2007, Neurosci Biobehav Rev 128:117-135, 2021). Previous studies indicated that the transcriptional activity of GR is not only regulated by the hormone levels. The GR phosphorylation status is also important in modulating the GR transcriptional activity (Mol Endocrinol 22:1331-1344, 2008). Under chronic stress, the GR gene was downregulated and the CHR gene was upregulated.

The change of b2-AR expression during chronic stress is similar to that of the GR (Psychoneuroendocrinology 99:191-195, 2019).

In this study, we found that chronic stress induced increased levels of both GR and b2-AR expression. Further studies are necessary to reveal the underlying mechanisms.

  • It is not clear whether chewing interferes with tumor implantation or tumor growth. The Authors report a decrease in the number of the lung metastatic nodules in the stress + chewing group as compared with the stress group. Another interesting parameter would have been nodule size. Based on the histograms in Figure 1B and 1C, it seems that the size of the nodules are in the same range in the stress group and in the stress + chewing group. In contrast, the histological sections seem to display higher nodule sizes in the stress group.

Response: In the present study, we found that the metastatic nodules were existed in mouse lung of all groups. However, the number and the size of the lung metastatic nodules in the stress group were significantly increased, compared with the control and stress with chewing groups. Therefore, we consider that chewing might attenuate stress-related tumor growth effects.

  • It is not clear how many sections were analyzed to determine the number of metastatic nodules. In the Material and method section, it is mentioned line 120 "n=6", does it mean 2 mice/group or 6 mice/group? And line 122: "Five fields were randomly selected from each section". Since panels 1B and 1C in Figure 1 display 5 dots/group, does it mean that for each group only one section from one mouse was analyzed? Due to the heterogeneity of tumor implantation, and the low number of metastatic nodules (mean value 1 to 4 nodules/group), I am not confident with the representativeness.

Response: As to determining the number of metastatic nodules under light microscope, we used 6 mice per group. Twenty sections per animal, 5 fields per section were randomly selected to determine the number and size of the lung metastatic nodules. We revised the paragraph in the Material and method section, 2.4. Histopathological image analysis, for easily understanding.

  • According to the Scale bars in Figure 1D, 1E ,1F, the metastatic nodules are smaller than 100um. So it is not clear to me, what is the "lung metastatic tissue" which was analyzed by western blot. Is it pools of nodules? in this case the Authors should describe the microdissection procedure. Is it homogenates from whole lungs? in this case this is not metastatic tissue. This should be clearly stated to delineate which cell types might produce VEGF, iNOS, VEGF, TGFB and MMPs (although there is probably stromal cells infiltrating the metastatic nodules). If whole lung have been homogenized, it would have been interesting to compare the expression of these proteins i) in another tissue, e.g. liver, and ii) in lung from control (non metastatic) mice subjected to a psychological stress.

Response: Thank you for this indication. It is possible to select the metastatic nodules using laser capture microdissection method (Methods Mol Biol. 755:245-56, 2011). In this study, we collected lung tissue homogenates for Western Blot analysis. Therefore, the phrase "lung metastatic tissue" is inappropriate, as you indicated. Therefore, the expression ‘Mouse lung metastatic tissue’ was changed to ‘Mouse lung homogenates’ in the Materials and Methods, and the Results sections. The whole lung tissue contains many kinds of cells, including cancer cells, infiltrating stromal cells, vascular endothelium, and the lung parenchymal cells. It is hard to state which cell types produce VEGF, iNOS, TNF-a, TGF-b and MMPs. Therefore, we added a paragraph to mention several potential limitations of this study in the Discussion section. As you suggested, we would like to investigate lung and liver metastasis in our next study.

  • From a general point of view, the Authors should provide some enlargements of the histological and immunohistological micrographs. The enlargement in Fig.1F seems different from Fig.1D and 1E.

Response: Thank you for your valuable comments. We added the enlargements in Fig. 1D, E, and F. We also exchanged the image of Fig.1F.

  • The original western blots (back matter) display a huge background, it is difficult to identify the main specific bands. This is also a problem for the interpretation of immunohistochemistry. As far as I understood, original western blots originate from 3 membranes. MMP9 (Fig.3E) and TGFB (Fig.3B) labeling have been performed sequentially on the same blot (regardless of the order), but the patterns are similar and thus the bands considered as specific are subject to caution. The expected size of B2-AR is higher than 46,000 (core protein), in the range 50,000-60,000 depending on glycosylation level. The immunoreactive band attributed to B2-AR in Fig.2D is probably below 40,000. Furthermore, its pattern of expression looks like a band of the same size on the iNOS blot (Fig.4A).  For TNF-a, another band of about 60,000 is also observed and displays the similar pattern of expression.

Response: We thank the reviewer for this important indication. Some western blots contain non-specific background signals due to our technical reasons. As you indicated, the patterns of MMP9 (Fig.3E) and TGF-b (Fig.3B) are similar, but the blot bands are different. As for b2-AR, the molecular weight is 46 kDa, as you indicated. However, the blot band appeared at around 38 kDa, not 46 kDa, the predicted band. Therefore, we performed the western blot again and found the ∼46-kDa band of b2-AR. Figure 2F for western blot results of b2AR was exchanged. The results showed that the expression b2AR in the stress group was significantly higher than that of the control and stress with chewing groups (p < 0.05). Please find the original western blot images of b2AR and GAPDH attached separately.

We do not know exactly the band of about 60,000 for TNF-a, the possibility of the existence of the TNF-a dimers and trimers can’t be excluded (Sci Rep 10:9065, 2020).

  • Glucococorticoids are potent immunomodulators. The Authors could consider in the future, alternative experimental models to immunodefficient mice, such as xenograft of mouse breast cancer 4T1 or EO771 cell lines in balb/c or C57BL/6 mice, respectively, or the use of transgenic mice or chemically induced carcinogenesis.

Response: Thank you for your kind suggestion. In the present study, we used MDA-MB-231 cell lines, the triple-negative breast cancer cell lines, and nude mice for examination. In the near future, we would like to apply MCF-7, T47-D, luminal A subtype of breast cancer cell lines, or 4T1, the mouse breast cancer cell line, and Balb/c, C57BL/6, or transgenic mice, to increase our understanding of the breast cancer and its metastasis.

Round 2

Reviewer 2 Report

Thank you for having taken into account my comments

Nevertheless, in line to my previous report, I am not confident with the western blot displayed. The original western blots (back matter) show a huge background, it is difficult to identify the main specific bands. The pattern of MMP9 (Fig.3E) and TGFB (Fig.3B) is superimposable (including the bands considered as specific, please see enclosed PDF file). Differences between blots might be attributed to the sequential antibody treatments. Even if the bands considered are specific, this overlay with non-specific signal impacts quantification. It should be stressed also that some nonspecific signals follow the same pattern of expression according to the groups of animals (stress > stress+chewing > control), and not always mimicked by GAPDH (e.g. TGF B Fig.3B).

Concerning, B2-AR, the Authors have performed a new western blot and found a band of 46kDa. Nevertheless, this blot still displays many other immunoreactive signals. As I previously mentioned, the core protein of B2-AR is indeed 46000, but the receptor is glycosylated leading to Mr higher than 50,000 - 60,000 according to cell / tissue types, depending on glycosylation level.

I would suggest that the Authors perform further experiments using new antibodies or after optimization of their conditions of western blot, and if possible including positive and negative controls.

Another point that remains unclear for me is the number of dots (n=5) in Figure 1 B & C since 6 mice were analyzed.

Author Response

Responses to the comments of Reviewer 2 

We thank this reviewer for the valuable comments. The response to the comments is as follows.

Nevertheless, in line to my previous report, I am not confident with the western blot displayed. The original western blots (back matter) show a huge background, it is difficult to identify the main specific bands.

The pattern of MMP9 (Fig.3E) and TGFB (Fig.3B) is superimposable (including the bands considered as specific, please see enclosed PDF file).

Differences between blots might be attributed to the sequential antibody treatments. Even if the bands considered are specific, this overlay with non-specific signal impacts quantification. It should be stressed also that some nonspecific signals follow the same pattern of expression according to the groups of animals (stress > stress+chewing > control), and not always mimicked by GAPDH (e.g. TGF B Fig.3B).

Concerning, B2-AR, the Authors have performed a new western blot and found a band of 46kDa. Nevertheless, this blot still displays many other immunoreactive signals.

As I previously mentioned, the core protein of B2-AR is indeed 46000, but the receptor is glycosylated leading to Mr higher than 50,000 - 60,000 according to cell / tissue types, depending on glycosylation level.

I would suggest that the Authors perform further experiments using new antibodies or after optimization of their conditions of western blot, and if possible including positive and negative controls.

Response: Because of our technical reasons, some western blots contain non-specific background signals. According to your suggestion, we performed the western blot for b2-AR, TGF-b, MMP9  and negative controls again.

As for b2-AR, we detected the band at approximately 60 kDa, coincident with the molecular weight of glycosalated b2-AR. We changed the figure 2F. The original western blot image was attached.

According to the instruction of the manufacturer (Cell Signaling Technology), the molecular weight of the monomer TGFb is 12 kDa and that of the mature or dimer TGFb is about 25 kDa. We detected the band of TGFb at approximately 25 kDa, corresponding to the instruction of the manufacturer. We changed the figure 3B. The original western blot image was attached.

According to the manufacturer’s instruction (Proteintech), the molecular weight of MMP9 is  67 kDa. We detected the band of MMP9 at 67 kDa, coincident with the instruction of the manufacturer. We changed the figure 3E. The original western blot image was attached.

Another point that remains unclear for me is the number of dots (n=5) in Figure 1 B & C since 6 mice were analyzed.

Response: Thank you for your indication. We used 18 mice (n=6) for this experiment. As to determining the number and size of the metastatic nodules under light microscope, we used 5 mice per group. Twenty sections per animal, 5 fields per section were randomly selected to determine the number and size of the lung metastatic nodules.

We made a correction from ‘n=6’ to ‘n=5’, in the Material and method section, 2.4. Histopathological image analysis, due to our mistake.

Round 3

Reviewer 2 Report

The Authors have tried to improve the immunoblots, but there is still a lot of background and the specificity of the selected bands remain questionable. Especially, there are no positive or negative controls

The reanalysis of the densitometry of B2AR did not evidence any statistical difference between the stress+chewing and the stress groups. This has been changed in the main text but not in the abstract. Please correct line 30.

As mentioned in a previous round of evaluation, the Authors did not analyze lung metastatic tissues, but lung homogenates. Please, correct this point in lines 32, 224, 228 and 241.

Author Response

Responses to the comments of Reviewer 2 

We thank this reviewer for the valuable comments. The response to the comments is as follows.

The Authors have tried to improve the immunoblots, but there is still a lot of background and the specificity of the selected bands remain questionable. Especially, there are no positive or negative controls.

The reanalysis of the densitometry of B2AR did not evidence any statistical difference between the stress+chewing and the stress groups. This has been changed in the main text but not in the abstract. Please correct line 30.

As mentioned in a previous round of evaluation, the Authors did not analyze lung metastatic tissues, but lung homogenates. Please, correct this point in lines 32, 224, 228 and 241.

Response: Thank you for your critical indication. We tried our best to improve the immunoblots manuscript by repeating the experiments.

According to your suggestion, we revised that sentence of line 30 of the Abstract as follows.

‘…attenuated the elevated expression of adrenergic receptors in lung tissues’, because the immunostaining results showed that b2AR-positive cells in the metastatic nodules of the stress group was significantly higher than stress with chewing groups. Western blot findings revealed that the expression of b2AR tended to be lower in stress with chewing group.

We also corrected the expression of lines 32, 224, 228 and 241, as you suggested.
